# Uterus-specific transcriptional regulation underlies eggshell pigment production in Japanese quail

Satoshi Ishishita[1], Shumpei Kitahara[2], Mayuko Takahashi[2], Sakura Iwasaki[2], Shoji Tatsumoto[3], Izumi Hara[4], Yoshiki Kaneko[2], Keiji Kinoshita[1], Katsushi Yamaguchi[5], Akihito Harada[6], Yasushige Ohmori[4], Yasuyuki Ohkawa[6], Yasuhiro Go[3,7], Shuji Shigenobu[5], Yoichi Matsuda[1,2]*, Takayuki Suzuki[1,2]*

1 Avian Bioscience Research Center, Graduate School of Bioagricultural Sciences, Nagoya University, Furo-cho, Chikusa-ku, Nagoya, Aichi, Japan, 2 Laboratory of Avian Bioscience, Department of Animal Sciences, Graduate School of Bioagricultural Sciences, Nagoya University, Furo-cho, Chikusa-ku, Nagoya, Aichi, Japan, 3 Cognitive Genomics Research Group, Exploratory Research Center on Life and Living Systems (ExCELLs), National Institutes of Natural Sciences, Okazaki, Aichi, Japan, 4 Laboratory of Animal Morphology, Department of Animal Sciences, Graduate School of Bioagricultural Sciences, Nagoya University, Furo-cho, Chikusa-ku, Nagoya, Aichi, Japan, 5 Functional Genomics Facility, National Institute for Basic Biology (NIBB), Okazaki, Aichi, Japan, 6 Division of Transcriptomics, Medical Institute of Bioregulation, Kyushu University, Higashi-ku, Fukuoka, Fukuoka, Japan, 7 Division of Behavioral Development, Department of System Neuroscience, National Institute for Physiological Sciences, Okazaki, Aichi, Japan

* suzuki.takayuki@j.mbox.nagoya-u.ac.jp (TS); yoimatsu@agr.nagoya-u.ac.jp (YM)

**Data Availability Statement:** The Illumina data generated in this study were deposited in the DDBJ Sequence Read Archive: ddRAD sequencing (Accession codes: DRA005969), mRNA

## Abstract

The precursor of heme, protoporphyrin IX (PPIX), accumulates abundantly in the uteri of birds, such as Japanese quail, *Coturnix japonica*, which has brown-speckled eggshells; however, the molecular basis of PPIX production in the uterus remains largely unknown. Here, we investigated the cause of low PPIX production in a classical Japanese quail mutant exhibiting white eggshells by comparing its gene expression in the uterus with that of the wild type using transcriptome analysis. We also performed genetic linkage analysis to identify the causative genomic region of the white eggshell phenotype. We found that 11 genes, including *5'-aminolevulinate synthase 1* (*ALAS1*) and *hephaestin-like 1* (*HEPHL1*), were specifically upregulated in the wild-type uterus and downregulated in the mutant. We mapped the 172 kb candidate genomic region on chromosome 6, which contains several genes, including a part of the *paired-like homeodomain 3* (*PITX3*), which encodes a transcription factor. *ALAS1*, *HEPHL1*, and *PITX3* were expressed in the apical cells of the luminal epithelium and lamina propria cells of the uterine mucosa of the wild-type quail, while their expression levels were downregulated in the cells of the mutant quail. Biochemical analysis using uterine homogenates indicated that the restricted availability of 5'-aminolevulinic acid is the main cause of low PPIX production. These results suggest that uterus-specific transcriptional regulation of heme-biosynthesis-related genes is an evolutionarily acquired mechanism of eggshell pigment production in Japanese quail. Based on these findings, we discussed the molecular basis of PPIX production in the uteri of Japanese quails.

sequencing (DRA006948), and whole-genome resequencing (DRA007010). All raw data are included in the Supporting information or deposited on GitHub (https://github.com/isst001/eggshell_color_gene).

**Funding:** This research was supported by the National BioResource Project (NBRP) Chicken/Quail of the Japan Agency for Medical Research and Development (https://www.amed.go.jp/en/index.html) and a Grant-in-Aid for Scientific Research on Innovative Areas from the Japan Society for the Promotion of Science (JSPS KAKENHI Grant Number JP23113004) (https://www.jsps.go.jp/english/index.html) to YM. The funders had no role in study design, data collection and analysis, decision to publish, or preparation of the manuscript.

**Competing interests:** The authors have declared that no competing interests exist.

## Introduction

Avian eggshells display diverse colors and patterns, which may be a result of adaptation to different habitats by birds. Eggshell colors have various functions, such as avoiding predation (camouflage) and promoting egg recognition by parents [1–3]. Elucidating the molecular and cellular basis for eggshell pigmentation is essential for understanding the evolutionary process and mechanism of eggshell coloration; however, it remains largely unknown [4–6]. Avian eggshell pigments are mainly composed of intermediates and/or catabolites of heme, including protoporphyrin IX (PPIX), an organic compound comprising four pyrrole rings, which is observed as a brown-colored pigment on the eggshell [7–9]. PPIX is the final intermediate in the eight-step heme biosynthesis pathway (S1 Fig), wherein the insertion of iron into PPIX generates heme, a component of hemoproteins [10]. PPIX levels are low in most tissues because heme biosynthesis is tightly regulated to avoid the toxic effect of PPIX accumulation [11], and the synthesized PPIX is efficiently converted to heme in the presence of ferrous iron. Exceptionally, PPIX accumulates in the uterus, especially in the caudal part of the oviduct, of bird species that produce brown eggs, indicating that the biosynthesis pathway of heme is specifically regulated in their uteri. The Japanese quail, *Coturnix japonica*, usually lays eggs with brown-speckled color patterns generated by PPIX. In the uteri of the birds of this species, a large amount of PPIX accumulates in the apical cells of the mucosal epithelium before its secretion [12, 13]. To elucidate the molecular basis of PPIX production in the uterus of Japanese quail, we focused on a classical quail mutant exhibiting white eggshells owing to the low PPIX production in its uterus [14, 15]. The cause of low PPIX production in this mutant quail remains unknown. We postulated that the genes involved in PPIX production should be specifically upregulated in the uterus of wild-type quail and downregulated in the uterus of the mutant quail exhibiting white eggshells. Therefore, in this study, we first compared the gene expression profiles of wild-type and mutant quail uteri using transcriptome analysis. Next, we performed genetic linkage mapping of the causative genomic region of the white eggshell phenotype and attempted to identify the candidate genes. The framework of this study in shown in S2 Fig.

## Materials and methods

### General

No statistical methods were used to predetermine the sample size, and the experiments were not randomized. The investigators were not blinded to the allocation during experiments and the outcome assessment.

### Ethics statement

Animal care and all experimental procedures were approved by the Animal Experiment Committee, Graduate School of Bioagricultural Sciences, Nagoya University (approval number: 2017030238). Experiments were conducted according to the Regulations on Animal Experiments in Nagoya University.

### Animals

To obtain the tissue, genomic DNA, and/or mRNA samples or construct a reference family for genetic linkage analysis, commercial female quails were purchased from a local hatchery (Cyubu-kagaku-shizai, Nagoya, Japan), and quail strains (WE, NIES-L, and NIES-Fr) [16–18] were supplied by the National BioResource Project Chicken/Quail, Nagoya University, Japan. The commercial quail and the NIES-L and NIES-Fr strains exhibited wild-type brown-

speckled eggshells, whereas the WE strain exhibited white eggshells. All quails were unvaccinated and maintained individually with free access to water and a commercially available diet. Most quails were maintained under a 14:10 h light/dark photoperiod; however, some were maintained under a 16:8 h light/dark photoperiod for use with quantitative polymerase chain reaction (qPCR) analysis. Room temperature (RT) was maintained at approximately 25°C. Egg-laying quails that were 16–48 weeks old were used for the animal experiments. Notably, a superficial layer of brown pigment is formed on the eggshell at approximately 21–22 h following the last oviposition in Japanese quail [19–21]. Oviducts were collected after decapitation of quail following isoflurane anesthesia. After all experiments, quail were decapitated after isoflurane anesthesia.

## mRNA sequencing and differential gene expression analysis

To identify the differentially expressed genes (DEGs) between the wild-type and mutant uteri and detect the genes that are specifically upregulated in the uterus (uterus-specific genes) compared to the other six organs (heart, liver, intestine, kidneys, lungs, and brain) and two tissues (muscle and isthmus), we performed mRNA sequencing of the uteri and isthmuses from the wild-type and mutant quail. For preparing tissue samples, females of both the commercial quail (exhibiting wild-type eggshells) and the WE mutant strain were used ($n$ = 6 for both). To obtain mucosal tissues from the isthmuses and uteri just before the secretion of eggshell pigments, female quails were sacrificed 19–20 h after the last oviposition. Then, the isthmuses and uteri were isolated from the oviducts and opened with scissors with the lumen side up. The tissues were rinsed with phosphate-buffered saline (PBS). Subsequently, ice-cold PBS containing 40 mM dithiothreitol was used as a buffer. Mucosal surfaces were scraped from the uteri and isthmuses by rubbing softly with an interdental brush, and the separated tissues were collected by centrifugation at $400 \times g$ and 4°C. Preliminary histological observation of the uteri and isthmuses after treatment confirmed that the epithelia were mainly separated from the mucosal folds (S3 Fig).

For the extraction of total RNA, 50 mg of the tissue pellet was dissolved in 1 mL of TRI Reagent (Molecular Research Center, Cincinnati, OH, USA) and total RNA was purified using the RNeasy Mini Kit (Qiagen, Hilden, Germany). RNA quality was assessed using Bioanalyzer Pico Chips (Agilent Technologies, Santa Clara, CA, USA). RNA samples with integrity numbers greater than 7.6 were used for constructing libraries. We isolated poly(A)+ RNA from 1 ng of total RNA using the NEBNext Poly(A) mRNA Magnetic Isolation Module (NEB, Ipswich, MA, USA), following the manufacturer's instructions. We converted oligo(dT)-selected RNA into a cDNA library for mRNA sequencing using the NEBNext Ultra Directional RNA Library Prep Kit for Illumina (NEB), according to the manufacturer's instructions. The library was sequenced on an Illumina HiSeq 1500 platform by paired-end sequencing (100 bp).

We trimmed the adapter sequences from the reads using Trimmomatic software v0.33 [22], and then mapped the reads to the reference genome (Accession code: GCF_001577835.1) using TopHat2 v2.1.0 [23]. Read counts per gene and fragments per kilobase of transcript per million mapped reads (FPKM) were calculated using Cuffdiff v2.2.1 [24] and Cufflinks v2.2.1 [25]. FPKM values were then converted into transcripts per kilobase million. We obtained the mRNA sequencing data of other organs and tissues of 15-month-old male quails from the Sequence Read Archive (Bio Project number: PRJNA296888), which were obtained by the analysis of six RNA samples for each organ or tissue on an Illumina HiSeq 2500 platform using paired-end sequencing (100 bp). The count data of the other organs and tissues were generated using the same procedure used in this study to obtain the mRNA sequencing data of

the uteri and isthmuses of quail. The average number of mapped read pairs is also shown in S1 Data.

We then compared the gene expression profiles between the uterus and other organs and tissues of the wild-type quail. Before detecting DEGs, we excluded genes from the dataset whose counts fell below the threshold (1) in any sample. Then, we performed likelihood ratio tests using the glmLRT function in EdgeR v3.20.9 [26]. In DEG analysis, the full model for count data was compared to the reduced model, in which the object coefficient was set to 0. $P$-values were adjusted for comparison of the expression levels between the uterus and each organ or tissue of quail using the Benjamini–Hochberg method. Genes whose counts were significantly larger (adjusted $p < 0.05$, fold change $> 2$) in the uterus were classified as genes that were specifically upregulated in the uterus. Hierarchical clustering was performed using the heatmap.2 function in the gplots package of R v.3.4.3 using Ward's method with the Euclidean distance metric.

## Construction of a reference family, double-digest restriction site-associated DNA (ddRAD) sequencing, and association test

For genetic linkage mapping of the causative genomic region, we constructed a reference family and obtained genome-wide single nucleotide polymorphism (SNP) markers by ddRAD sequencing, and then performed an association test using these SNP markers and the eggshell color phenotype data. To construct the reference family for genetic linkage analysis, one male of the WE strain and one female of the NIES-L strain that exhibited wild-type eggshells were used as parents. The $F_2$ hybrids were generated by crossing an $F_1$ hybrid male with three $F_1$ hybrid females. Genomic DNA was extracted from the red blood cells of the parents and 99 $F_2$ hybrid females using the DNeasy Blood & Tissue Kit (Qiagen). The library was constructed according to a previously described method [27]. After digesting 100 ng of DNA from each sample using EcoRI and MseI (NEB), adapters containing barcode sequences were ligated to the ends of the DNA fragments. DNA fragments ranging from 300–500 bp were collected using Pippin Prep (Sage Science, Beverly, MA, USA) and purified using AMPure XP beads (Beckman, Brea, CA, USA). The purified fragments were amplified (six cycles) with PCR primer sets containing index sequences using Phusion High-Fidelity DNA polymerase (NEB). Library quality was validated using 2100 Bioanalyzer with an Agilent High-Sensitivity DNA Kit (Agilent Technologies). The library was sequenced on an Illumina HiSeq 1500 platform by paired-end sequencing (100 bp) at the Functional Genomics Facility, National Institute for Basic Biology, Japan.

Sequence data were demultiplexed using their barcodes, cleaned by removing the low quality reads, and trimmed to 95 or 100 bp using Stacks v1.44 [28]. The processed reads were aligned to the reference sequence using Bowtie 2 v2.2.8 [29]. SNPs were assigned by aligning paired-end reads to the reference sequence using Stacks. SNPs with a depth of coverage of less than 8× were counted as missing data. After eliminating SNPs that deviated from the Hardy–Weinberg equilibrium ($p < 0.000001$) or had call rates of less than 80%, an association test was performed using Fisher's exact test implemented in PLINK v1.90 [30]. Sequences of the adapters and primers are shown in S1 Data.

## Whole-genome resequencing

To obtain SNP markers and candidate mutations around the locus responsible for the white eggshell phenotype, we performed whole-genome resequencing of the parent quails used to generate $F_1$ hybrids in genetic linkage analysis, as described above. Genomic DNA was extracted from the red blood cells of the parent quails. Library construction and paired-end

sequencing were performed using an Illumina HiSeq X Ten platform (150 bp) at BGI Shen-zhen. After trimming the reads using Trimmomatic, the reads were mapped to the reference sequence (Accession code: GCF_001577835.1) using BWA-MEM v0.7.15-r1140 [31]. Subsequently, variants on chromosome 6 were called using Picard v1.119 (http://broadinstitute.github.io/picard/) and the Genome Analysis Toolkit [32], according to a variant call pipeline (https://github.com/gencorefacility/variant-calling-pipeline). We filtered out bases with low Phred-scaled quality scores (less than 30) and reads with low mapping quality (less than 30) using the Genome Analysis Toolkit. Genotypes with individual coverage of less than 10× or more than 70× were filtered out using SnpSift v4.3t [33], and sequence variants that were found in the candidate causative region (candidate region) were annotated using SnpEff v4.3t [33]. To narrow down the candidate region, we searched for candidate genomic intervals for informative SNP markers that could reveal the parental origins of chromosomal regions of the $F_2$ progeny. To identify causative sequence variants around the candidate region, we extracted sequence variants that were homozygous in the WE strain and then filtered out those found in the NIES-L strain. We searched for structural variants around the candidate region using Integrative Genomics Viewer v2.6.3 [34] and Pindel v0.2.5b9 [35]. Nucleotide sequences were determined using an ABI PRISM 3130 DNA Analyzer (Applied Biosystems, Foster City, CA, USA) after cycle-sequencing reactions using the BigDye Terminator v1.1 Cycle Sequencing Kit (Applied Biosystems). Primer sequences are listed in S1 Table.

## qPCR analysis

To compare the gene expression levels of target genes between the wild-type and mutant uteri, qPCR analysis was performed using RNA samples, which were used for mRNA sequencing ($n$ = 5 for both the commercial quail and WE strain). qPCR analysis was also performed using RNA samples extracted from the uterine tissues of female quail that were not used for mRNA sequencing ($n$ = 5 for both the commercial quail and WE strain). Tissue collection and RNA extraction for these additional 10 RNA samples were conducted using the same method as that used for mRNA sequencing; however, quails were maintained under a 16:8 h light/dark photoperiod, and uterine tissues were collected 17 h after oviposition [36]. Total RNA (500 ng) was reverse transcribed into cDNA in a 10 μL reaction mixture using oligo(dT) and the ReverTra Ace qPCR RT Master Mix with a gDNA Remover (Toyobo, Osaka, Japan). PCR amplification (0.1 μL cDNA in a 10 μL reaction mixture) was performed using the Thunderbird SYBR qPCR Mix (Toyobo) and the StepOnePlus Real-Time PCR System (Applied Biosystems, Carlsbad, CA, USA). Melt curve analysis was performed immediately after the amplification. β-actin was used as an endogenous control. Each experiment was performed in triplicate. For all primer sets, the amplification efficiency was 92–112%, and the correlation coefficient was greater than 0.97. The primer sequences and PCR conditions are listed in S1 Table.

## Histology

To investigate the accumulation of PPIX in the uterine mucosa of wild-type and mutant quails, uteri were collected from female quails ($n$ = 6 for both the commercial quail and WE strain) 19–20 h after oviposition and fixed overnight in Bouin's solution. The fixed tissues were dehydrated in a series of graded ethanol, immersed in a 1:1 ethanol/xylene solution, and subsequently embedded in paraffin. The tissues were sectioned to a thickness of 4 μm and mounted onto ovalbumin-coated glass slides. After deparaffinization, the sections were stained with Mayer's hematoxylin and eosin or left unstained.

## Transmission electron microscopy (TEM)

Ultrastructural analyses of the mucosal epithelia of the wild-type and mutant uteri were performed using TEM to investigate the distribution of transport vesicles in apical cells of the uterine mucosa. Uterine samples were collected from female quails ($n = 1$ for both the commercial quail and WE strain) 19–20 h after oviposition, fixed overnight in 2% glutaraldehyde (in 0.1 M phosphate buffer) at 4˚C, rinsed with 0.1 M phosphate buffer overnight at 4˚C, and post-fixed in 2% osmium tetroxide (in deionized water [DW]) for 2 h at 4˚C. The fixed specimens were dehydrated in a series of graded ethanol (50, 70, 90, 100, 100, and 100%) for 15 min at RT, immersed in propylene oxide for 30 min, followed by immersion in a propylene oxide and epoxy resin mixture for 2 h at RT, and finally embedded in gelatin capsules with epoxy resin for two days at 60˚C. Ultrathin sections of 80 nm thickness were cut using an ultramicrotome with diamond knives and mounted on copper grids with a mesh size of 200 μm. The sections were stained with 2% uranyl acetate (in DW) for 15 min and lead staining solution for 5 min at RT. TEM was performed using an H-7600 TEM (Hitachi, Tokyo, Japan) operated at an accelerating voltage of 100 kV.

## *In situ* hybridization

To investigate the distribution of the transcripts of target genes in the uterine mucosa of wild-type and mutant quails, *in situ* hybridization analyses were performed. For the synthesis of RNA probes, partial sequences of cDNAs of the target genes were obtained by PCR amplification (see S1 Table for primer sequences and S2 Data for partial sequences of cDNAs) and then inserted into the pGEM-T Easy Vector System (Promega, Madison, WI, USA). RNA probes were synthesized *in vitro* using SP6 RNA polymerase and plasmid DNA digested with SphI (*5'-aminolevulinate synthase 1* (*ALAS1*)) and ApaI (*hephaestin-like 1* (*HEPHL1*) and *paired-like homeodomain 3* (*PITX3*)). Uterine tissues were collected from female quails ($n = 2$ for both the commercial quail and the WE strain) at 19–20 h after ovulation, fixed with 4% paraformaldehyde (PFA) overnight, and dehydrated sequentially with 25, 50, and 75% MeOH/PBT (PBS, 0.1% Tween 20), and 100% MeOH for 5 min each. After dehydration, the tissues were immersed in 100% EtOH and 100% xylene for 1 h each. Subsequently, the tissues were embedded in Tissue-Tek Paraffin WaxII60 (Sakura, Tokyo, Japan) and 10 μm sections were prepared using a Leica RM2125 microtome (Leica, Wetzlar, Germany). Sections were dried on glass slides overnight and then hydrated sequentially with 100% xylene, 100% EtOH, 90% EtOH, 70% EtOH, and PBT for 5 min each. After the hydration process, the slides were immersed in 1 μg/mL proteinase K in PBT for 7 min at 37˚C. Slides were washed thrice with PBT for 5 min and then fixed with 4% PFA for 20 min. The sections were hybridized with 1 μg/mL of DIG-labeled RNA in the hybridization buffer (50% formamide, 5× saline-sodium citrate [SSC], 1 mg/mL total RNA, 100 μg/mL heparin, 0.1% Tween 20, 0.1% 3-[(3-cholamidopropyl)dimethylammonio]-1-propanesulfonate [CHAPS], and 10 mM ethylenediaminetetraacetic acid [EDTA]) overnight at 68˚C. The slides were washed twice with 50% formamide in 2× SSC for 30 min. After cooling, the slides were incubated with the anti-DIG antibody (1:1,000; Roche, Penzberg, Germany) overnight. Then, the slides were washed thrice with Tris-buffered saline with 0.1% Tween 20 for 5 min and the color was developed in AP buffer (100 mM Tris-HCl, 100 mM NaCl, 10 mM $MgCl_2$, 4.5 μL/mL NBT [Roche], and 3.5 μL/mL BCIP [Roche], pH 9.5).

## Gene Ontology (GO) term enrichment analyses

To infer biological processes that are involved in eggshell pigment production and those affected in the mutant uterus, GO term enrichment analyses were performed using the

overrepresentation test (released 2021-02-24) of the Protein Analysis Through Evolutionary Relationships (PANTHER) Classification System (annotation version 16.0 released 2020-12-01) [37]. The *Gallus gallus* database was used as the reference. PANTHER GO-Slim Biological Process was used as the annotation dataset. *P*-values of Fisher's exact test were adjusted using the Benjamini–Hochberg method. GO terms were considered significantly enriched if they had a false discovery rate of < 0.05.

### Biochemical assay to evaluate the PPIX-forming ability of the uterus

To test whether the insufficiency of 5'-aminolevulinic acid (ALA) or its precursor causes low PPIX production, we compared the PPIX-forming abilities of the uterine homogenates of wild-type and mutant quails in the presence of high concentrations of ALA. Uterine tissues were collected from female quails (*n* = 1 for both the commercial quail and WE strain) 15–20 h after oviposition and homogenized in ice-cold PBS containing 0.05% Triton-X100. After centrifugation at $20,000 \times g$ and 4°C for 5 min, the supernatants of each homogenate were incubated in the presence of 0, 1, or 2 mM ALA (Wako, Osaka, Japan) in a 96-well plastic plate at 39°C for 20 h (three reactions were performed for each concentration of ALA in 110 μL homogenates). The absorbance was measured at 405 nm using an ARVOx4 2030 Multilabel Reader (PerkinElmer, Waltham, MA, USA) because PPIX exhibits maximum light absorbance at 409 nm. Supernatants incubated without ALA supplementation were used as controls. The total protein concentrations of the supernatants were determined using the Bradford method, and the mean absorbance per 1 mg/mL total protein concentration was calculated for each sample at each ALA concentration.

### Statistical analysis

R v.3.4.3 (R Core Team) and MS Excel (Microsoft Corp., Redmond, WA, USA) were used for all statistical analyses. For the comparison of expression levels by qPCR analysis, Welch's two-tailed *t*-test was performed using delta *Ct* values. Welch's two-tailed *t*-test was also used to test the differences in the absorbance (at 405 nm) of uterine homogenates between the wild-type and mutant quails. The significance (alpha) level of the statistical tests was set to 0.05.

## Results

### Histological analysis of the uterine mucosae of the wild-type and mutant quails

We performed gross and histological examination of the uterine mucosae of wild-type and mutant quails, which lay eggs with normal and white eggshells, respectively (Fig 1A; S4 Fig). The uterine mucosa of the wild-type quail exhibited a dark brown color, and that of the mutant quail was pale brown (Fig 1B). Histological examination revealed that brown pigment granules were accumulated in the apical cells of the mucosal epithelium of the uterus in the wild-type quail before anticipated secretion (19–20 h after oviposition), whereas few brown pigment granules were observed in the mutant quail (Fig 1C and 1D). Instead, pink-colored, eosinophilic granules accumulated in the cytoplasm of apical cells in the mutant, suggesting that transport vesicles containing few or no PPIX were formed even in the mutant cells. Electron microscopic observation of the uterine mucosa sections also showed that transport vesicles were accumulated in the apical cells of both the wild type and mutant (S5 Fig).

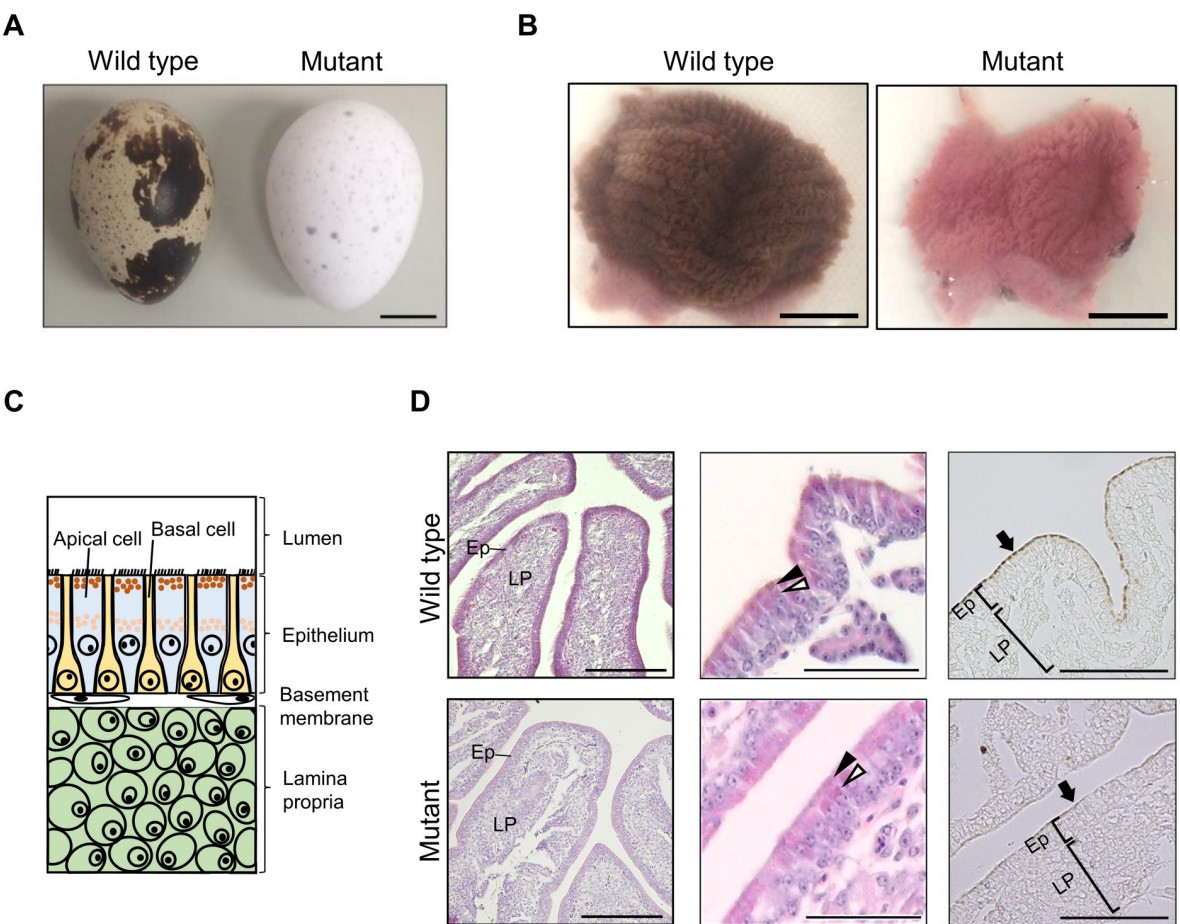

**Fig 1. White eggshell phenotype and low accumulation of brown pigments in the uterine mucosa of the mutant quail.** (A) Wild-type and mutant white eggs. (B) Wild-type and mutant uteri were opened with the lumen side up. (C) Schematic diagram of the uterine mucosa. Epithelium of the uterine mucosa consists of alternatingly arranged apical and basal cells. Pigment granules containing protoporphyrin IX (PPIX) molecules accumulate in the luminal side of the cytoplasm of apical cells before secretion. Lamina propria is located on the inner side of the uterine mucosa. (D) Sections of uterine mucosae obtained from wild-type and mutant quails just before pigment secretion (left and middle panels, HE staining; right panels, no staining). Ep and LP indicate epithelium and lamina propria, respectively. Brown pigments accumulated in the epithelium of the uterine mucosa of the wild-type quail, but not in that of the mutant quail (left panels). Higher-magnification images of mucosal folds (middle panels) indicate that brown pigment granules accumulated in the wild-type apical cells, but not in the mutant apical cells (black arrowheads); however, pink-colored, eosinophilic granules accumulated in the luminal side of the mutant apical cells. Eosinophilic granules also accumulated more centrally in both the wild-type and mutant apical cells (white arrowheads). Sections without staining clearly demonstrate that brown pigments were accumulated in the luminal side of the apical cells (arrows) of the wild-type quail, but not in the apical cells of the mutant quail. Scale bars: 1 cm in (A, B), 100 μm in (D).

## Characterization of the mutant uterus by gene expression analysis and identification of uterus-specific genes

To characterize the mutant uterus at the gene expression level, we performed mRNA sequencing analysis using the uterine tissues of wild-type and mutant quail and compared gene expression profiles between the wild-type and mutant uteri. The results showed that 261 out of approximately 17,000 expressed genes were differentially expressed (adjusted $p < 0.05$, fold change $< 0.5$ or $> 2$) (Fig 2A; S3 Data). These DEGs consisted of 148 and 113 genes that showed upregulated and downregulated expression, respectively, in the mutant (Fig 2A; S3 Data). We then focused on changes in the expression levels of genes known to be involved in heme and iron metabolism [11] and found that *ALAS1* [38], *HEPHL1* [39], and *transferrin*

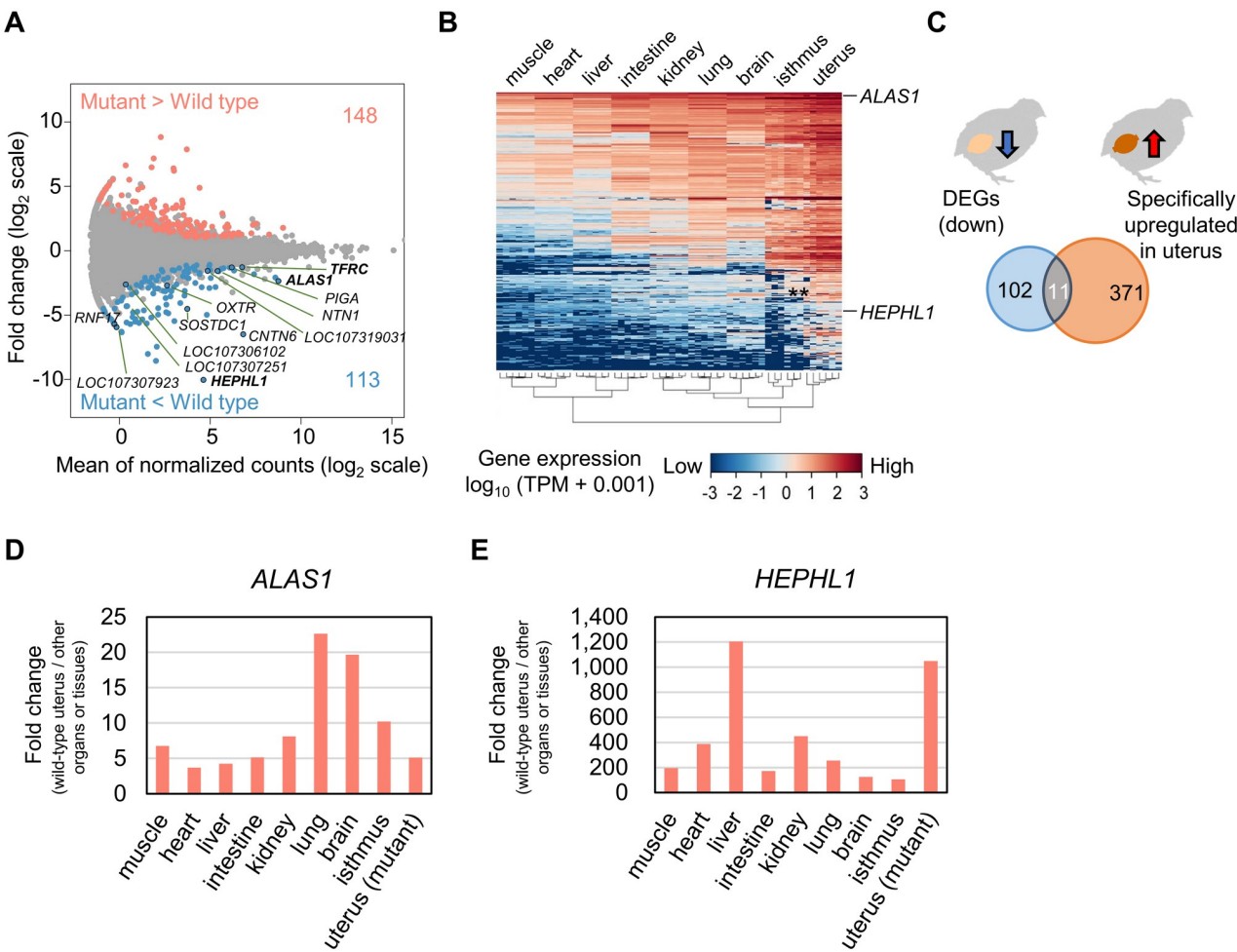

**Fig 2. Identification of genes that are specifically upregulated in the wild-type uterus and downregulated in the mutant uterus.** (A) Visualization of the result of differential gene expression analysis using wild-type and mutant uterine tissues. Dots indicate the expression levels of genes. Differentially expressed genes (DEGs) [adjusted $p < 0.05$, fold change ($\log_2$ scale) $> 1$ or $< -1$] are highlighted. Genes that were downregulated in the mutant and specifically upregulated in the wild-type uteri and *transferrin receptor protein 1* (*TFRC*) are labeled. (B) Gene expression levels of the 382 genes whose expression levels were higher in the uterus than in the other six organs and two tissues, visualized by log-transformed transcripts per kilobase million (TPM) values. (C) The number of downregulated genes in the mutant uterus, uterus-specific genes in the wild type, and genes showing both expression patterns. (D, E) Comparison of the expression levels of *5'-aminolevulinate synthase 1* (*ALAS1*) (D) and *hephaestin-like 1* (*HEPHL1*) (E) in the wild-type uterus with those in other wild-type organs and tissues and the mutant uterus, using mRNA sequencing data. Each bar indicates the relative value of the expression level of the gene in the wild-type uterus when the expression level in each organ or tissue and the mutant uterus is defined as 1. Values in (D, E) are not indicated by the log scale.

receptor protein 1 (*TFRC*) [40] were downregulated in the mutant uterus (Fig 2A; S2 Table). The expression levels of these genes in the mutant decreased to 20% (*ALAS1*), 0.1% (*HEPHL1*), and 41% (*TFRC*) of those in the wild type (S2 Table).

To identify uterus-specific genes, we compared gene expression profiles between the uterus and the other six organs and two tissues. The results showed that 382 genes were specifically upregulated in the uterus (Fig 2B; S4 Data). The expression levels of genes related to heme and iron metabolism in the wild-type uterus were compared to those in other organs and tissues (S3 Table). Eleven of the DEGs, including *ALAS1* and *HEPHL1*, were also included in the uterus-specific genes (Fig 2C; S3 and S4 Tables; S3 and S4 Data). The expression levels of *ALAS1* and *HEPHL1* were 4–22 times and 171–1,205 times higher, respectively, in the uterus

than in other organs and tissues (Fig 2D and 2E; S4 Table). Significant downregulation of *ALAS1* and *HEPHL1* in the mutant uterus was confirmed by qPCR analysis (Welch's two-sided *t*-test, $p < 0.01$) (S6 Fig; S5–S11 Data). The results of GO term enrichment analyses of the uterus-specific genes and DEGs are shown in S5 and S6 Tables, respectively. GO "biological process" terms, such as extracellular matrix organization and cation homeostasis, were overrepresented in both groups.

## Genetic linkage mapping of the responsible locus

The white eggshell phenotype of the mutant quail is inherited in an autosomal recessive manner [14]. However, the causative mutation, which we refer to as *white eggshell* (*we*), remains unknown. Therefore, to identify this mutation, we performed chromosomal mapping of the locus by genetic linkage analysis using 99 $F_2$ progeny obtained by mating three pairs of $F_1$ progeny derived from crossing one mutant male (the WE strain) with one wild-type female (the NIES-L strain). SNP markers were obtained by ddRAD sequencing of the genomic DNA of the 99 $F_2$ progeny and their parents. The average number of aligned read pairs and phenotype data are shown in S1 Data. An association test using a total of 15,200 SNP markers revealed a significant association of 235 SNP markers on chromosome 6 (Fisher's exact test, adjusted $p < 0.01$) with the white eggshell phenotype (Fig 3A; S12 Data). The marker SNP201288 displayed the highest association (Fig 3A) and logarithm of the odds (LOD) score and the lowest recombination frequency (Fig 3B and 3C; S12 Data); its genotypes in the $F_2$ progeny were fully concordant with the eggshell color phenotype (Fig 3D). Two SNP markers, SNP201255 (located adjacent to SNP201288) and SNP202246, were separated by a large genomic interval (2,230 kb) (Fig 3E). SNP202246 was located far (2,179 kb) from SNP201288 and displayed discordance between genotypes and phenotypes in eight $F_2$ progeny (Fig 3D).

To narrow down the candidate region, we carried out whole-genome resequencing of the parental quail, which was used for genetic linkage analysis, and obtained sequence variant information on chromosome 6. Sequence data of 33 and 45 GB were obtained for the father (WE strain) and mother quail (NIES-L strain), respectively (S1 Data). The average depth of coverage of chromosome 6 was approximately 16 and 23 for the father and mother quails, respectively (S1 Data). Structural variants, including copy number variation, large deletions or insertions, and inversions, could not be found around the candidate region. Almost all informative SNP markers were detected in the vicinity of SNP201288 and SNP202246 (S13 Data). Among them, the genotypes of Chr6-we1 and Chr6-we2 were concordant and discordant, respectively, with the eggshell color phenotype of the $F_2$ progeny (#4040) (Fig 3F; S7 Fig; S7 Table). Furthermore, Chr6-we-a, which was detected in a 2,030 kb genomic region between Chr6-we1 and Chr6-we2, displayed a concordance between the genotypes and phenotypes of the eight $F_2$ progeny (S7 Fig; S7 Table). However, the genotype of Chr6-we-a in the mutant was also observed in the quail strain NIES-Fr, which exhibited wild-type eggshells (S7 Table). Therefore, we eliminated the 2,058 kb region between Chr6-we1 and SNP202246 from the 2,230 kb candidate region, narrowing down the candidate region to the remaining 172 kb region located between SNP201255 and Chr6-we1.

## Identification of the causative mutation and causative gene around the *we* locus

To identify the candidate mutation, we searched for sequence variants (SNPs and insertion–deletion mutations [indels]) that were homozygous in the father quail (mutant) but not found in the mother quail (wild type). The results showed that 957 candidate sequence variants were detected in the 172 kb candidate region (Fig 4A and 4B; S8 Table; S13 Data). The 172 kb

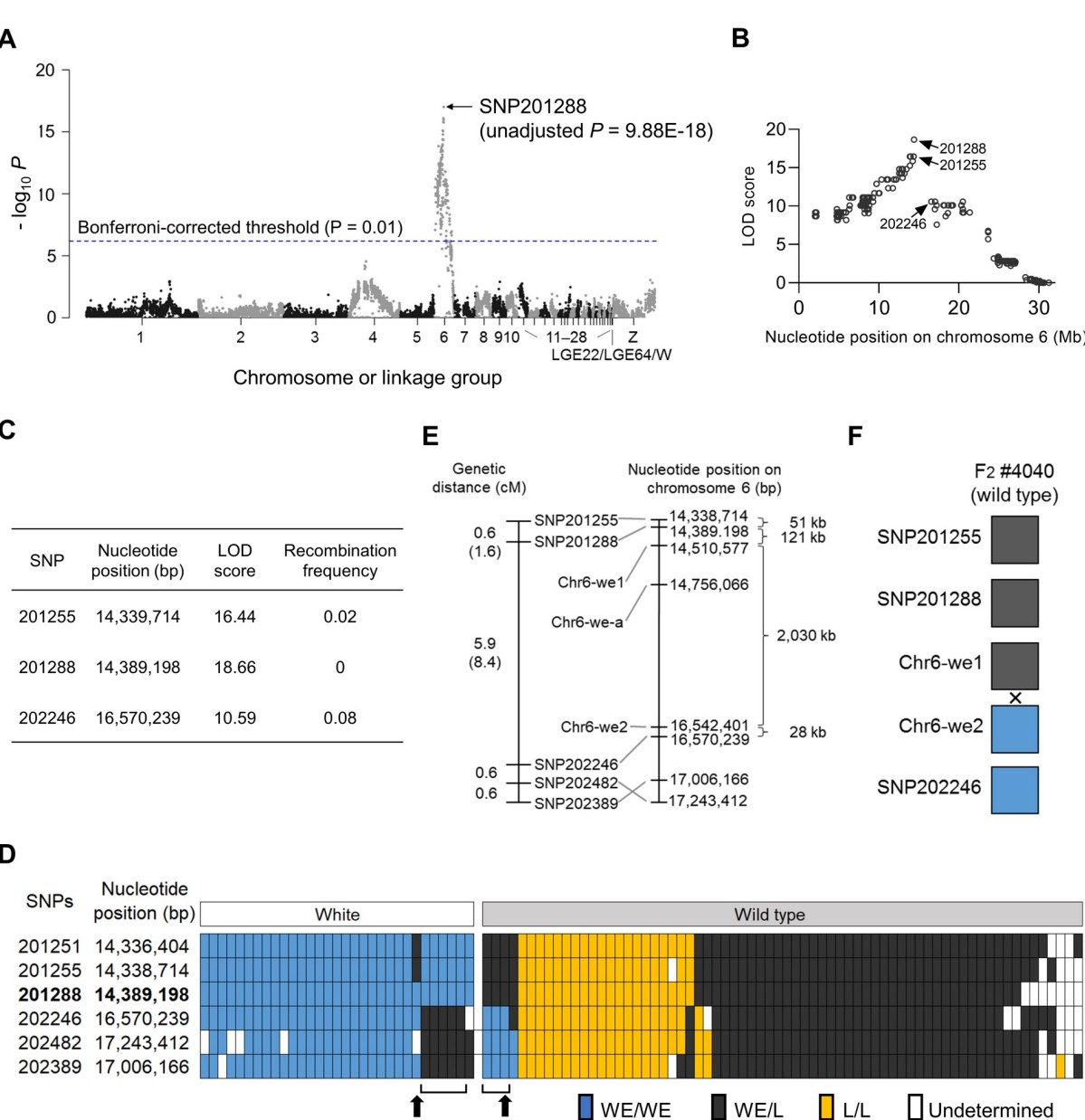

**Fig 3. Genetic linkage mapping of the *we* locus.** (A) Association test using 15,200 single nucleotide polymorphism (SNP) markers. *x*-axis, genomic coordinates of SNPs; *y*-axis, negative logarithm of *p*-values from Fisher's exact test. Blue line indicates the Bonferroni-corrected *p*-value of 0.01. SNP201288 displayed the strongest association with the phenotype. (B) Logarithm of the odds (LOD) scores between the *we* locus and all SNP markers on chromosome 6. SNP201288 and flanking SNPs (SNP201255 and SNP202246) display high LOD scores. (C) LOD scores and recombination frequencies of three SNPs near the *we* locus. (D) Genotypes of SNPs near the *we* locus in 99 $F_2$ individuals. SNP201288 shows a complete concordance between the genotypes and eggshell colors in the $F_2$ progeny. Discordance between the genotypes and eggshell colors was observed for SNP201255 in $F_2$ individual #3398 and for SNP202246 in eight $F_2$ progeny shown by brackets, including the $F_2$ individual #4040. (E) Genetic and physical maps of SNP markers around the *we* locus. Genetic distances between the *we* locus and flanking markers are indicated by parentheses. Positions of SNP202482 and SNP202389 are inverted between the genetic and physical maps, possibly due to an assembly error in the draft genome. (F) Genotypes of Chr6-we1 and Chr-we2 are concordant and discordant, respectively, with the eggshell color, in the $F_2$ individual #4040. Recombination in the interval between SNP markers is indicated by "×".

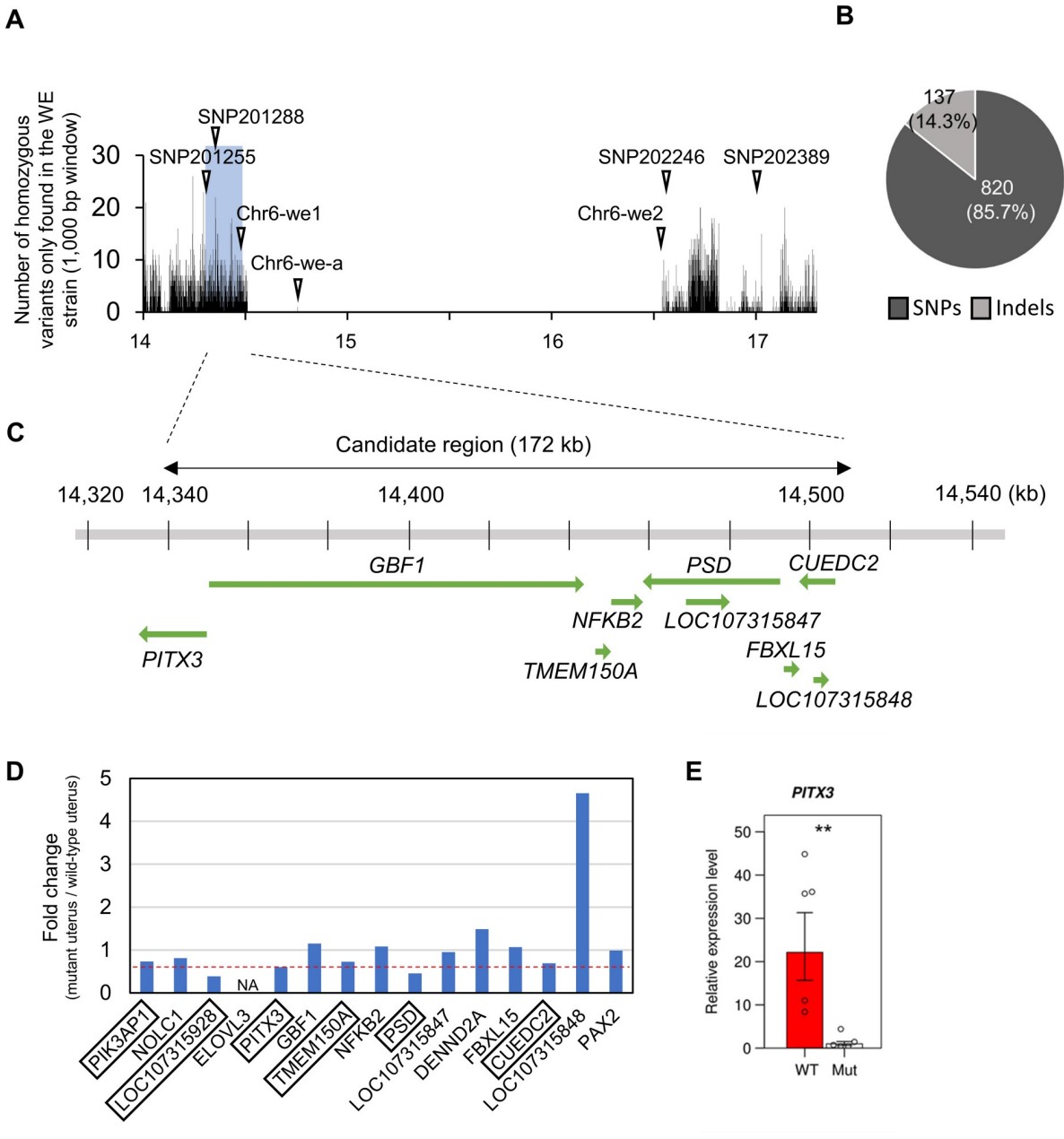

**Fig 4. Distribution of sequence variants around the *we* locus and expression analysis of genes around the 172 kb candidate region.** (A) Number of sequence variants that were homozygous in the father quail (WE strain), but not in the mother quail (NIES-L strain), around the *we* locus. White arrowheads indicate the locations of the SNP markers. (B) Frequencies of SNPs and insertion–deletion mutations (indels) within the 172 kb candidate region. (C) Eight genes and a part of *paired-like homeodomain 3* (*PITX3*) that are harbored in the 172 kb candidate region (green arrows). (D) Relative expression levels of 15 genes (located around the 172 kb candidate region) in the mutant uteri against those in the wild-type uteri. NA denotes genes whose expression was not detected in the wild-type uteri. Expression levels of six genes (enclosed in squares) substantially decreased in the mutant uteri (less than 60–70% of those in the wild-type uteri). (E) Graphs showing the results of qPCR analysis of *PITX3* using RNA samples (*n* = 5 for both wild-type and mutant quails) that were not used for mRNA sequencing. The expression levels in the wild-type uterus are shown relative to those in the mutant uterus defined as 1. The expression level in each sample is shown by a circle. **p < 0.01.

candidate region harbored seven protein-coding genes (*PITX3*, *GBF1*, *TMEM150A*, *NFKB2*, *PSD*, *FBXL15*, and *CUEDC2*) and two non-coding genes (*LOC107315847* and *LOC107315848*) (Fig 4C). Six missense variants were included in the 957 sequence variants (S8 Table; sheets 5 and 6 in S13 Data); however, none were predicted to have adverse effects on the functions of the protein-coding genes, including *GBF1*, *NFKB2*, *PSD*, and *CUEDC2* (S9 Table). Furthermore, although it is difficult to precisely estimate the effect of sequence variants, the remaining sequence variants were not considered to be related to functions (sheet 6 in the S13 Data). These results raised the possibility that the mutation affects the expression level of the gene, rather than its function.

Therefore, we predicted that the recessive mutation suppresses the expression of the responsible gene, because mutations that enhance gene expression levels are most likely to be dominant and not recessive. Therefore, we searched for genes on chromosome 6 that were downregulated in the mutant uteri using our mRNA sequencing data. Two genes, *PRKG1* and *C6H10orf71*, were significantly downregulated in the mutant (adjusted $p < 0.05$, fold change $< \frac{1}{2}$) (S10 Table; sheet 5 in S3 Data). However, they do not seem to be causative genes because both are located far ($> \sim 1$ Mb) from the 172 kb candidate region. Thus, we focused on the genes located within the 172 kb candidate region and in the region up to 100 kb on either side (nucleotide positions 14.23–14.61 Mb), of which six genes (*PIK3AP1*, *LOC107315928* [*2-hydroxyacylsphingosine 1-beta-galactosyltransferase-like*], *PITX3*, *TMEM150A*, *PSD*, and *CUEDC2*), were substantially downregulated in the mutant (their expression was less than approximately 70% of that in the wild type) (Fig 4D). *PITX3* encodes a transcription factor involved in the normal development, differentiation, and maintenance of cells in other tissues [41–44]. The other five genes encode proteins that may function in molecular and cellular processes other than transcriptional regulation, such as galactosylceramide biosynthesis and signal transduction [45–49]. Because *PITX3* is likely involved in a tissue-specific transcriptional regulation process, we expected that this gene could be a strong candidate. qPCR analysis using mRNA samples that were used for mRNA sequencing also showed non-significant but substantial downregulation of *PITX3* in the mutant uterus (S8 Fig; S5 and S14 Data). We observed significant downregulation of *PITX3* in the mutant uterus by qPCR analysis using additional RNA samples (Welch's two-sided *t*-test, $p < 0.01$) (Fig 4E; S5 and S15 Data). The difference between the qPCR analyses may be due to the unstable expression of *PITX3*. However, these results consistently indicate that *PITX3* is downregulated in the mutant.

## Distribution analysis of *PITX3*, *ALAS1*, and *HEPHL1* transcripts in the uterine mucosa

Subsequently, to identify tissues that produce PPIX, we focused on two uterus-specific genes, *ALAS1* and *HEPHL1*, and *PITX3*, and examined the expression patterns of these genes in sections of the uterine mucosa of wild-type and mutant quail. We investigated the distribution of *PITX3*, *ALAS1*, and *HEPHL1* transcripts in the uterine mucosa of wild-type and mutant quail. These three genes were expressed in apical cells of the mucosal epithelium and cells of the lamina propria (LP), the latter of which are located adjacent to the basal layer of the mucosal epithelium, in the wild type (upper panels of Fig 5). In contrast, hybridization signals were very weak for all three genes in both apical cells and LP cells in the mutant (lower panels of Fig 5). Notably, the hybridization signals of *ALAS1* were more intense in the LP cells than in the apical cells of the wild-type quail (Fig 5A). These results raise the possibility that the mucosal epithelium and lamina propria are involved in PPIX production.

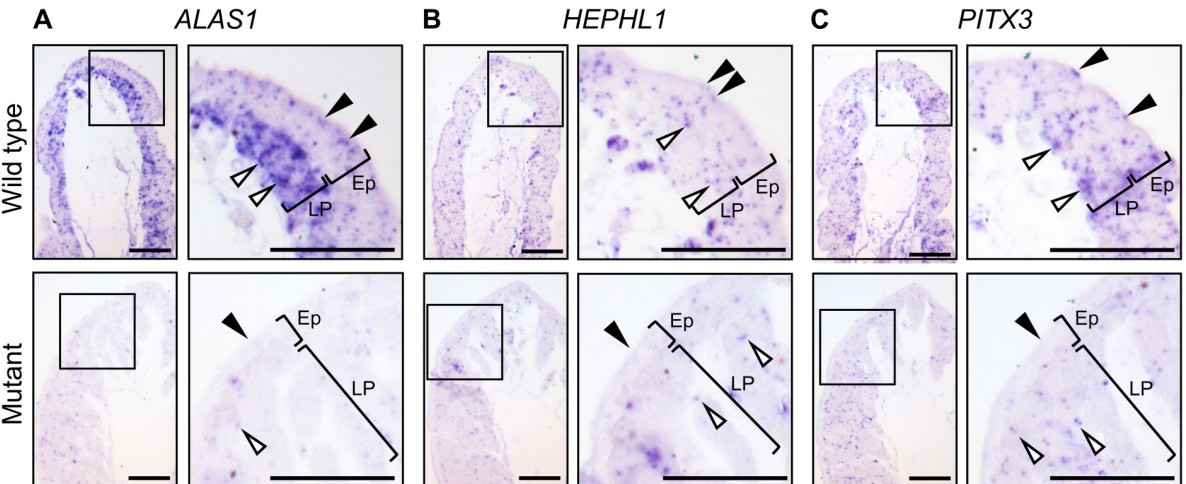

**Fig 5. Distribution of *PITX3*, *ALAS1*, and *HEPHL1* transcripts in uterine mucosal tissues.** (A–C) Light micrographs of the sections of mucosal folds of the wild-type (upper panels) and mutant (lower panels) uteri. *PITX*, *ALAS1*, and *HEPHL1* were expressed in the apical cells on the luminal side of the epithelia (Ep) of the wild-type uteri, and their expression was observed as bluish hybridization signals (upper panels, black arrowheads). Hybridization signals were also observed in the cells of the lamina propria (LP) (upper panels, white arrowheads). *ALAS1* was highly expressed in the cells of LP. The hybridization signals of three genes in the luminal side of the Ep (lower panels, black arrowheads) and LP cells (lower panels, white arrowheads) of the mutant were weak compared to those of the wild type.

## Comparison of the PPIX-forming abilities between the wild-type and mutant uterine homogenates

Finally, to test the causal relationship between low *ALAS1* expression and low PPIX production in the mutant uterus, we examined whether the PPIX-forming ability in uterine homogenates of the mutant quail could be recovered by supplementation with a large amount of ALA. The results showed that homogenates from mutant uteri could produce the same amount of PPIX as those from wild-type uteri in both 1 mM and 2 mM concentrations of ALA (Welch's two-sided *t*-test, $P \geq 0.05$) (Fig 6; S16 Data), indicating that the restricted availability of ALA or its precursors due to downregulated *ALAS1* expression is the main cause of the low PPIX production.

## Discussion

Eggshell PPIX is synthesized in the uterus and accumulates in the mucosal epithelium of birds; however, where and how PPIX is synthesized in the uterus remains largely unknown. In this study, we investigated the cause of low PPIX production in a Japanese quail mutant that laid white eggshells by transcriptome and genetic linkage analyses.

Transcriptome analysis revealed that the expression levels of 11 genes were higher in the uterus than in other organs and tissues in the wild-type quail and were downregulated in the mutant uterus. Therefore, we expected that these genes would be involved in the uterus-specific regulation of heme synthesis, which should be disrupted in the mutant quail. Among these 11 genes, *ALAS1* and *HEPHL1*, which are well-known genes related to heme and iron metabolism, were included. *ALAS1* encodes the rate-limiting enzyme of heme biosynthesis, which catalyzes ALA synthesis as the first step in heme biosynthesis. Our results indicate that high *ALAS1* expression in the uterus is required for the increased synthesis of PPIX [38, 50]. The PPIX-forming ability was comparable between the wild-type and mutant uteri in the presence of large amounts of ALA, which suggests that the restricted availability of ALA, owing to

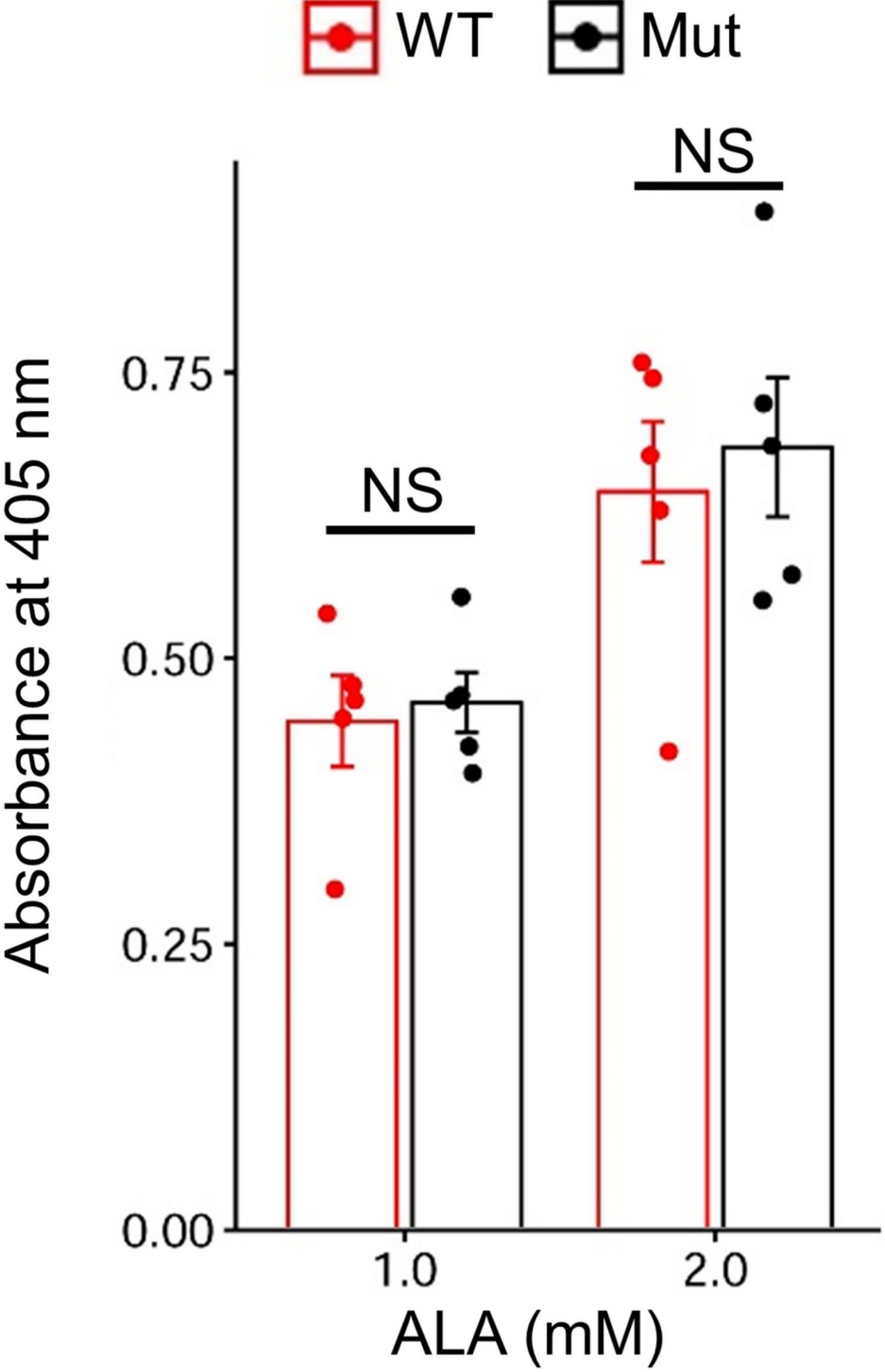

**Fig 6. Comparison of the PPIX-forming abilities of the uterine homogenates between the wild-type and mutant quails.** Absorbance of uterine homogenates incubated in the presence of different concentrations of 5'-aminolevulinic acid (ALA). Bars indicate the mean absorbance at 405 nm per 1 mg/mL total protein. Dots indicate the absorbance of each sample. Error bars indicate the standard error of the mean.

the downregulation of *ALAS1*, is the main cause of low PPIX production in the mutant quail. However, a previous biochemical study showed that the ability to synthesize PPIX was lower in mutant homogenates than in wild-type homogenates [15]. The difference in results between this study and previous studies may be due to the differences in the genetic background of the mutant and/or wild-type quail used in the experiments. A previous biochemical study using chickens also suggested that limited availability of ALA or its precursor in the uterus could be the cause of white eggshells [51]. *ALAS1* expression in the uterus was lower in a chicken strain that laid lighter brown eggshells than in those with darker brown eggshells [52]. These results and our present data suggest that low ALA synthesis, due to the downregulation of *ALAS1*, may be a genetic cause of white eggshells in both quail and chickens. However, we cannot exclude the possibility that there are additional factors that affect PPIX production, other than an insufficient amount of ALA *in vivo*. *HEPHL1* encodes a copper-dependent ferroxidase that converts ferrous iron to ferric iron [39]. Thus, HEPHL1 is likely involved in PPIX accumulation by inhibiting the conversion of PPIX to heme through its ferroxidase activity. Therefore, downregulation of *HEPHL1* is another possible cause of low PPIX production in the mutant quail. Although the remaining nine genes, except for *ALAS1* and *HEPHL1*, have not been shown to be related to heme and iron metabolism (S4 Table), these genes are also important in elucidating the molecular basis of PPIX production in the uterus.

The candidate locus responsible for white eggshells was localized to a 172 kb region on chromosome 6, which contains nine genes, by genetic linkage analysis. Although many candidate mutations were found within this region, no sequence variants were expected to influence gene function. Several genes that are located up to 100 kb on either side of this 172 kb region, including *PITX3*, were substantially downregulated in the mutant. Because *PITX3* is likely involved in a tissue-specific transcriptional regulation process [41–44], it is plausible that *ALAS1* expression is regulated by PITX3. Therefore, downregulation of *PITX3* is one possible cause of low PPIX production in the mutant. Further investigations, such as identification of the causative mutation and transgenic rescue experiments, are required to verify this possibility. The causative mutation may affect *PITX3* expression via changes in histone modification of enhancer chromatins. Epigenetic approaches, such as chromatin immunoprecipitation sequencing with anti-H3K27ac antibody [53] or assay for transposase-accessible chromatin using sequencing (ATAC-seq) [54], to compare active regulatory regions in uterine cells between the wild type and mutant would be useful for identifying candidates of uterus-specific enhancers of *PITX3*.

Based on our findings, we propose a hypothetical model of the uterus-specific regulation of heme biosynthesis and the underlying transcriptional regulation, as shown in Fig 7. Heme biosynthesis actively progresses with abundant ALA synthesis through the upregulation of *ALAS1*. In parallel with this, the conversion of PPIX to heme may be inhibited through a ferroxidase reaction mediated by the predicted ferroxidase HEPHL1. This inhibition of the final step of heme biosynthesis results in the accumulation of a large amount of PPIX in the uterus. Transcription factors, such as PITX3, may be involved in the transcriptional regulation of these two genes. This model requires the production of a large amount of glycine and succinyl-CoA, which are precursors of ALA in uterine cells. However, to verify our hypothesis, the downstream targets of PITX3 need to be identified. In addition, there may be other possible mechanisms of eggshell pigment production, such as bird-uterus-specific translation and post-translational regulation.

This study demonstrated that *ALAS1* is highly expressed in the apical cells of the epithelium and lamina propria (LP) cells of the uterine mucosa, indicating the possible involvement of these cells in PPIX synthesis through ALA synthesis. This concurs with the suggestion that the amount of PPIX produced is too large to be produced only in the apical cells of the uterine

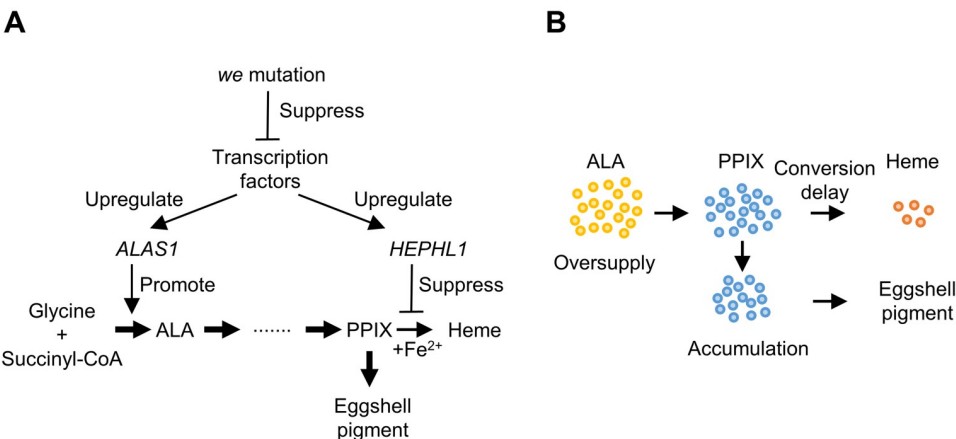

**Fig 7. Hypothetical model of PPIX production in the uterus of Japanese quail.** (A) Hypothetical model of the uterus-specific regulation of heme biosynthesis and underlying transcriptional regulation. Transcription factors may increase the expression levels of *ALAS1* and *HEPHL1* in the uterus. The upregulated expression of *ALAS1* may enhance ALA synthesis, and facilitate the excessive production of PPIX by uterine cells, leading to its accumulation by exceeding the conversion of PPIX to heme. In addition, HEPHL1, a predicted ferroxidase, may suppress the conversion of PPIX to heme by decreasing the intracellular or intramitochondrial ferrous ions, which may accelerate the accumulation of PPIX. (B) Schematic diagram of PPIX accumulation in the uterine cells.

mucosa [55]. However, gene expression analysis of each cell type in the uterine mucosa should be conducted to clarify the role of the LP and apical cells in PPIX production.

Many quantitative trait loci that control eggshell color in chickens have been reported [5]. In addition, correlations between the depth of brown eggshell color and the expression levels of genes involved in heme biosynthesis have been reported in chickens [52]. However, no causative genes for eggshell brownness/whiteness in chickens have been identified. *SLCO1B3*, encoding an anion transporter, is the first and only gene that has been identified as a causative gene for eggshell color variation; the overexpression of this gene due to retrovirus insertion leads to a blue egg phenotype in chickens [56]. The findings of this study may help to identify the causative gene of white eggshells, contributing to a deeper understanding of the genetic basis of eggshell color variation.

## Conclusions

In this study, we postulated that uterus-specific transcriptional regulation plays an important role in the uterus-specific regulation of the heme biosynthesis pathway and that this specific transcriptional regulation mechanism is disrupted in the white eggshell mutant quail. Based on this postulation, we performed transcriptome analysis of the wild-type and mutant uteri as well as genetic linkage mapping of the genomic region responsible for white eggshells. The results showed that 11 genes were specifically upregulated in the wild-type uterus and downregulated in the mutant uterus. Among them, *ALAS1* and *HEPHL1* were highly expressed in the apical cells of the mucosal epithelium and LP cells of the wild-type uterus, but downregulated in the mutant uterus. The results from our biochemical analysis indicated that ALA insufficiency was mainly responsible for the low PPIX production in the mutant. Furthermore, we showed that several genes in the 172 kb candidate region, including the gene encoding the transcription factor PITX3, were downregulated in the mutant uterus. These results collectively suggest that tissue-specific transcriptional regulation underlies PPIX production in the uterus. Although further investigation is required to verify our model, our findings may aid in

uncovering the physiological mechanisms of PPIX production in the bird uterus as well as the genetic basis of eggshell color variation. In future studies, we should identify the causative gene and mutation responsible for white eggshells and investigate the functional roles and transcriptional mechanisms of the 11 genes that were specifically expressed in the wild-type uterus and downregulated in the mutant uterus. In addition, the specific roles of apical and LP cells in PPIX production should be further elucidated.

## Supporting information

**S1 Fig. Schematic diagram of the heme metabolism pathway.** The image in the upper panel was prepared by modifying the figure from Sachar et al. (2016) [11].
(TIF)

**S2 Fig. Framework of this study.**
(TIF)

**S3 Fig. Mucosae of the wild-type uteri after brushing and mucosae of isthmuses before and after brushing.** (A–D) Mucosae in uteri after brushing. (B, D) Higher-magnification images of the parts shown inside the frames in (A, C). The epithelium is removed from the mucosa (arrows). Arrowheads indicate the remaining epithelia. (E–H) Mucosae of the isthmuses before brushing (E, F) and after brushing (G, H). (F, H) Higher-magnification images of the parts shown inside the frames in (E, G). Scale bar indicates 100 μm.
(TIF)

**S4 Fig. External shape of quail oviducts.** Schematic diagram of the quail oviducts containing an egg (A). The uteri of the wild-type and mutant quails were dark brown and pale brown, respectively (B). The vaginas were removed from the oviducts. Scale bar indicates 1 cm.
(TIF)

**S5 Fig. Transmission electron microscopy of the uterine mucosal epithelium.** (A, B) Wild-type uterus. (C, D) Mutant uterus. The part shown inside the frame in (A) is enlarged in (C). Nuclei of apical and basal cells are indicated by arrows and arrowheads, respectively. Transport vesicles are observed as electron-dense granules in the apical cells of both the wild-type and mutant epithelia (asterisks). Magnifications are 4000× in (A), 10000× in (B), 1500× in (C), and 7000× in (D).
(TIF)

**S6 Fig. Comparison of the expression levels of *ALAS1* and *HEPHL1* between the wild-type and mutant uteri by quantitative polymerase chain reaction (qPCR).** The graphs show the results of qPCR analysis using RNA samples ($n = 5$ for both wild-type and mutant quails) used for mRNA sequencing (A, B) and RNA samples ($n = 5$ for both wild-type and mutant quails) that were not used for mRNA sequencing (C, D). The expression levels in the wild-type uterus are shown as relative values to those in the mutant uterus, defined as 1. The expression level in each sample is shown as a circle. $^{**}p < 0.01$.
(TIF)

**S7 Fig. Genotypes of single nucleotide polymorphisms (SNPs) located near the candidate region in nine $F_2$ progeny that exhibited discordance between phenotypes and genotypes of SNP201255 or SNP202246.** Genotypes of SNPs 201288, Chr6-we1, and Chr6-we-a were fully concordant with the eggshell color phenotypes.
(TIF)

**S8 Fig. Comparison of *PITX3* expression levels between wild-type and mutant uteri by qPCR.** The graphs show the results of qPCR analysis using RNA samples ($n = 5$ for both wild-type and mutant quails) used for mRNA sequencing. The expression levels in the wild-type uterus are shown as relative values to those of the mutant uterus, defined as 1. The expression level in each sample is shown as a circle. NS, not significant. **$p < 0.01$.
(TIF)

**S1 Data. Statistics of mRNA, double-digest restriction site-associated DNA (ddRAD), and whole-genome resequencing; information on DNA samples for genetic linkage analysis; and nucleotide sequences of the adaptors and primers for the construction of ddRAD sequencing libraries.**
(XLSX)

**S2 Data. Sequences of cDNAs used for *in situ* hybridization.**
(DOCX)

**S3 Data. Raw count data used in the comparison of the gene expression profiles between the wild-type and mutant uteri and the result of differential gene expression analysis.**
(XLSX)

**S4 Data. Raw count data used in the comparison of the gene expression profiles between the uteri and other organs or tissues and the result of differential gene expression analysis.**
(XLSX)

**S5 Data. Results of the quantitative polymerase chain reaction (qPCR) analysis of *5'-aminolevulinate synthase 1* (*ALAS1*), *hephaestin-like 1* (*HEPHL1*), and *paired-like homeodomain 3* (*PITX3*).**
(XLSX)

**S6 Data. Raw data from real-time PCR analysis of *ALAS1* expression (experiment 1–1).** Data from three samples from both wild-type and mutant quails that were used for mRNA sequencing.
(XLSX)

**S7 Data. Raw data from real-time PCR analysis of *ALAS1* expression (experiment 1–2).** Data from two samples from both wild-type and mutant quails that were used for mRNA sequencing.
(XLSX)

**S8 Data. Raw data from real-time PCR analysis of *ALAS1* expression (experiment 2).** Data of five samples from both wild-type and mutant quails that were not used for mRNA sequencing.
(XLSX)

**S9 Data. Raw data from real-time PCR analysis of *HEPHL1* expression (experiment 1).** Data from five samples from both wild-type and mutant quails that were used for mRNA sequencing.
(XLSX)

**S10 Data. Raw data from real-time PCR analysis of *HEPHL1* expression (experiment 2–1).** Data of five samples from wild-type quails that were not used for mRNA sequencing.
(XLSX)

**S11 Data. Raw data from real-time PCR analysis of *HEPHL1* expression (experiment 2–2).** Data of five samples from mutant quails that were not used for mRNA sequencing.
(XLSX)

**S12 Data. Results from the genome-wide association test, logarithm of the odds (LOD) scores, and recombination frequencies of single nucleotide polymorphisms (SNPs) on chromosome 6.**
(XLSX)

**S13 Data. Sequence variants around the *we* locus and annotation of candidate sequence variants within the candidate region.**
(XLSX)

**S14 Data. Raw data from real-time PCR analysis of *PITX3* expression (experiment 1).** Data from five samples from both wild-type and mutant quails that were used for mRNA sequencing.
(XLSX)

**S15 Data. Raw data from real-time PCR analysis of *PITX3* expression (experiment 2).** Data of five samples from both wild-type and mutant quails that were not used for mRNA sequencing.
(XLSX)

**S16 Data. Results from biochemical analysis.**
(XLSX)

# Acknowledgments

We would like to thank Hisayo Asao and Asaka Akita (NIBB), Kayako Fukuyama (Kyushu University), Makoto Mizutani, Mikiharu Nakano, and Mitsuo Nunome (Nagoya University) for their technical assistance. The quail strains used in this study were provided by the National BioResource Project (NBRP) Chicken/Quail, supported by the Ministry of Education, Culture, Sports, Science, and Technology (MEXT) and Japan Agency for Medical Research and Development (AMED), Japan. ddRAD sequencing was carried out under the NIBB Cooperative Research Program for Next-generation DNA Sequencing (15–833). Computations were performed on the NIG supercomputer at the National Institute of Genetics.

# Author Contributions

**Conceptualization:** Satoshi Ishishita, Shoji Tatsumoto, Keiji Kinoshita, Katsushi Yamaguchi, Yasuhiro Go, Shuji Shigenobu, Yoichi Matsuda.

**Formal analysis:** Satoshi Ishishita, Shumpei Kitahara, Mayuko Takahashi.

**Funding acquisition:** Yoichi Matsuda.

**Investigation:** Satoshi Ishishita, Shumpei Kitahara, Mayuko Takahashi, Sakura Iwasaki, Shoji Tatsumoto, Izumi Hara, Katsushi Yamaguchi, Akihito Harada, Yasushige Ohmori, Yasuyuki Ohkawa, Yasuhiro Go, Shuji Shigenobu, Takayuki Suzuki.

**Methodology:** Satoshi Ishishita, Shumpei Kitahara.

**Project administration:** Satoshi Ishishita.

**Resources:** Keiji Kinoshita, Katsushi Yamaguchi, Yasuyuki Ohkawa, Yasuhiro Go, Shuji Shigenobu, Yoichi Matsuda, Takayuki Suzuki.

**Software:** Satoshi Ishishita, Shoji Tatsumoto, Akihito Harada.

**Supervision:** Yoichi Matsuda, Takayuki Suzuki.

**Validation:** Satoshi Ishishita, Shoji Tatsumoto, Katsushi Yamaguchi, Yasuhiro Go, Shuji Shigenobu.

**Visualization:** Satoshi Ishishita, Shumpei Kitahara, Mayuko Takahashi, Yoshiki Kaneko.

**Writing – original draft:** Satoshi Ishishita.

**Writing – review & editing:** Satoshi Ishishita, Yoichi Matsuda, Takayuki Suzuki.

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
