## [Decision Letter · Decision Letter 0]

30 Dec 2021

PONE-D-21-31931Uterus-specific transcriptional regulation underlies eggshell pigment production in Japanese quailPLOS ONE

Dear Dr. Suzuki,

Thank you for submitting your manuscript to PLOS ONE. After careful consideration, we feel that it has merit but does not fully meet PLOS ONE’s publication criteria as it currently stands. Therefore, we invite you to submit a revised version of the manuscript that addresses the points raised during the review process.

ACADEMIC EDITOR: Minor revision. 

We look forward to receiving your revised manuscript.

Kind regards,

Karthikeyan Adhimoolam

Academic Editor

PLOS ONE

Journal Requirements:

Reviewers' comments:

Reviewer's Responses to Questions

**Comments to the Author**

1. Is the manuscript technically sound, and do the data support the conclusions?

Reviewer #1: Yes

Reviewer #2: Yes

2. Has the statistical analysis been performed appropriately and rigorously? 

Reviewer #1: Yes

Reviewer #2: Yes

3. Have the authors made all data underlying the findings in their manuscript fully available?

Reviewer #1: Yes

Reviewer #2: Yes

4. Is the manuscript presented in an intelligible fashion and written in standard English?

Reviewer #1: Yes

Reviewer #2: Yes

5. Review Comments to the Author

Reviewer #1: In this manuscript entitled “Uterus-specific transcriptional regulation underlies eggshell pigment production in Japanese quail”, the author has mainly demonstrated a causative gene of white eggshells in Japanese quail by the approach of mRNA sequencing, ddRAD sequencing, whole-genome resequencing. In Summary, the author in this manuscript have clearly demonstrated the transcriptomic profiling to uncover the genes responsible for white eggshells. The manuscript can be accepted without any further revision.

Reviewer #2: The manuscript entitled “Uterus-specific transcriptional regulation underlies eggshell

pigment production in Japanese quail”. The authors analysed the cause of low PPIX production in a classical Japanese quail mutant exhibiting white eggshells by comparing its gene expression in the uterus with that of the wild type using transcriptome analysis and performed genetic linkage mapping to identify the causative genomic region of the white eggshell phenotype. It also suggests that uterus-specific transcriptional regulation of heme-biosynthesis-related genes is an evolutionarily acquired mechanism of eggshell pigment production in Japanese quail. Based on our findings, we discuss the molecular basis of PPIX production in the uterus of Japanese quail.

The manuscript comprises all the necessary elements of scientific novelty. The experimental designing and execution of the study were appreciable. I recommend this manuscript for the reconsideration for publication in PlosOne after incorporating minor changes given in below.

COMMENTS FOR THE AUTHOR:

Authors must concentrate on the formatting, and use of symbols, etc., throughout the manuscript.

Framework figure is required. It will be useful to the readers for better understanding of the studied issue.

In materials and method section: authors should explain why each item of methodology was done.

Proper formatting needed in the materials and methods, results, discussion and conclusion section.

Introduction was comprehensive to the studied issue.

Real Time-PCR’s raw data in the supplementary for scrutiny.

In conclusion section authors should provide some future prospectus related to present study.

6. PLOS authors have the option to publish the peer review history of their article (what does this mean?). If published, this will include your full peer review and any attached files.

Reviewer #1: No

Reviewer #2: No

---

## [Author Response · Author response to Decision Letter 0]

15 Feb 2022

Response to Reviewer #2

We wish to express our appreciation to the Reviewer for his or her insightful comments, which have helped us significantly improve the paper.

1) “Authors must concentrate on the formatting, and use of symbols, etc., throughout the manuscript.”, “Proper formatting needed in the materials and methods, results, discussion and conclusion section.”

Response:

We have revised the formatting properly for readability. The descriptions in the Materials and methods, results, and discussion sections have been arranged. Please see the revised manuscript with track changes for details. Main revisions are as follows.

Lines 140. “The average number of mapped read pairs is also shown in S1 Data.”, which was placed in the result section in the original manuscript, has been moved to the Materials and Methods section, for readability.

Lines 250-256. “For the synthesis of RNA� paired-like homeodomain 3 (PITX3))”, which was placed at the end of this paragraph, has been moved next to the first sentence of the paragraph for readability.

Line 469. The line break position has been changed. In the original manuscript, it was placed before “The 172 kb candidate region harbored�” (line 443 in the revised manuscript).

Lines 566-591. Discussion about the candidate gene and the hypothetical model has been moved from the 4th and 5th paragraphs in the original manuscript to the 3rd and 4th paragraphs in the revised manuscript.

Lines 603-608. The discussion about in situ hybridization analysis has been moved from the 3rd paragraph to the 5th paragraph.

Consequently, the numbering of reference has been changed as follows.

#53 in the original manuscript > #55 in the revised manuscript

#54 > #53

#55 > #54

The reference list has been changed along with these alterations.

The result of biochemical assay and the hypothetical model of the genetic cascade in the original Fig 6 have been divided into Fig 6 and Fig 7 in the revised manuscript, for readability.

We have defined “uterus-specific genes” and “the candidate region” as follows for readability.

Lines 106-110.

“To identify the differentially expressed genes (DEGs) between the wild-type and mutant uteri and detect the genes that are specifically upregulated in the uterus (uterus-specific genes) compared to the other six organs (heart, liver, intestine, kidneys, lungs, and brain) and two tissues (muscle and isthmus), we performed mRNA sequencing of the uteri and isthmuses from the wild-type and mutant quail.”

Line 193-195.

“sequence variants that were found in the candidate causative region (candidate region) were annotated using SnpEff v4.3t [33].”

In the author contribution section (at the end of the manuscript), the short names of several authors have been changed, as follows.

Lines 917-923. 

Ishishita > Is

Iwasaki > Iw

Ohkawa > Ohk

Ohmori > Ohm

We have arranged or added several subtitles in Materials and Methods and Result sections for readability, as follows.

Line 105. “mRNA sequencing” has been change to “mRNA sequencing and differential gene expression analysis”.

Line 152-153. “Double-digest restriction site-associated DNA (ddRAD) sequencing and association test” has been changed to “Construction of a reference family, double-digest restriction site-associated DNA (ddRAD) sequencing, and association test”.

Line 232. “Transmission electron microscopy” has been changed to “Transmission electron microscopy (TEM)”.

Line 286. “Biochemical assay for the PPIX-forming ability of the uterus” has been changed to “Biochemical assay to evaluate the PPIX-forming ability of the uterus”.

Line 323-324. “Fig 1. The white eggshell phenotype” has been changed to “Fig 1. White eggshell phenotype and low accumulation of brown pigments in the uterine mucosa of the mutant quail.” for scientific accuracy.

Line 342-343. “Characterization of the mutant uterus by gene expression analysis” has been changed to “Characterization of the mutant uterus by gene expression analysis and identification of uterus-specific genes”.

Line 356-357. “Fig 2. Genes that are specifically upregulated in the wild-type uterus and downregulated in the mutant uterus.” has been changed to “Fig 2. Identification of genes that are specifically upregulated in the wild-type uterus and downregulated in the mutant uterus.”.

Line 386. “Genetic linkage mapping of the candidate region responsible for the white eggshell phenotype” has been changed to “Genetic linkage mapping of the responsible locus”.

Line 404. “Fig 3. The candidate region of the white eggshell (we) phenotype is mapped to chromosome 6.” has been changed to “Fig 3. Genetic linkage mapping of the we locus.”.

Line 440-441. “Identification of the causative mutation and causative gene around the we locus” has been added.

Lines 494-495. “Distribution analysis of PITX3, ALAS1, and HEPHL1 transcripts in the uterine mucosa” has been added.

2) ” Framework figure is required.”

Response:

We have added framework of this study in S2 Fig, therefore,

Line 77-78. We have mentioned “The framework of this study in shown in S2 Fig.” at the end of Introduction.

Line 814. We have added a figure caption “S2 Fig. Framework of this study.”. 

The numbering of the remaining Supporting Figures has also been changed.

3) ”In materials and method section: authors should explain why each item of methodology was done.”

Response:

Lines 91-92, 106-110, 154-157, 181-183, 224, 233-235, 249-250, 277-278. We have described the objective of each item at the beginning of paragraphs. Please read the revised manuscript with track changes for details.

We have also mentioned objectives of experiments in the result section at lines 343-345, 389, 442, and 496. 

4) ”Real Time-PCR's raw data in the supplementary for scrutiny.”

Response:

As suggested, we have added raw data of Real Time PCRs as S6�S11, S14, and S15 Data.

Because we have added these data, numbering of Supporting Data and has been changed and captions in pages 39�41 have been changed.

5) ”In conclusion section authors should provide some future prospectus related to present study.”

Response:

We have added the following sentences at the end of the conclusion.

Lines 635-640. “In future studies, we should identify the causative gene and mutation responsible for white eggshells and investigate the functional roles and transcriptional mechanisms of the 11 genes that were specifically expressed in the wild-type uterus and downregulated in the mutant uterus. In addition, the specific roles of apical and LP cells in PPIX production should be further elucidated.”

Journal Requirements:

1. Please ensure that your manuscript meets PLOS ONE's style requirements, including those for file naming. The PLOS ONE style templates can be found at https://journals.plos.org/plosone/s/file?id=wjVg/PLOSOne_formatting_sample_main_body.pdf

and https://journals.plos.org/plosone/s/file?id=ba62/PLOSOne_formatting_sample_title_authors_affiliations.pdf

We have corrected the manuscript style.

We have uploaded all data and procedures to replicate our study onto PLOS ONE submission system or public database. Therefore, we would like to remove the following statement “Other miscellaneous information is available from the corresponding author upon request.”

Our data deposited to public database (SRA in DDBJ) and github.com are already accessible, without restriction.

There are no papers that have been retraced in our reference list. 

We have mentioned about changes to the reference in rebuttal letter.

While revising your submission, please upload your figure files to the Preflight Analysis and Conversion Engine (PACE) digital diagnostic tool, https://pacev2.apexcovantage.com/. PACE helps ensure that figures meet PLOS requirements. To use PACE, you must first register as a user. Registration is free. Then, login and navigate to the UPLOAD tab, where you will find detailed instructions on how to use the tool. If you encounter any issues or have any questions when using PACE, please email PLOS at figures@plos.org. Please note that Supporting Information files

do not need this step.

We have checked all the figures using the PACE tool. Revised figures (Figs 2, 6, and 7; S2 Fig) have been corrected via PACE.

5. Others

The following careless mistakes have been corrected as follows.

Line 431.

Fig 3E (at line 452 in the original manuscript) > Fig 3F

In Fig 2C.

The number of DEG (down) in the circle has been corrected from 250 (all 261 DEGs – 11 overlapping genes) to 102 (=113 downregulated genes – 11 overlapping genes).

Many minor corrections in terms of grammatical accuracy have been made though professional English proofreading. See the revised manuscript with track changes for details.

---

## [Decision Letter · Decision Letter 1]

22 Feb 2022

Uterus-specific transcriptional regulation underlies eggshell pigment production in Japanese quail

PONE-D-21-31931R1

Dear Dr. Suzuki,

We’re pleased to inform you that your manuscript has been judged scientifically suitable for publication and will be formally accepted for publication once it meets all outstanding technical requirements.

Kind regards,

Karthikeyan Adhimoolam

Academic Editor

PLOS ONE

Additional Editor Comments (optional):

Reviewers' comments:

Reviewer's Responses to Questions

**Comments to the Author**

1. If the authors have adequately addressed your comments raised in a previous round of review and you feel that this manuscript is now acceptable for publication, you may indicate that here to bypass the “Comments to the Author” section, enter your conflict of interest statement in the “Confidential to Editor” section, and submit your "Accept" recommendation.

Reviewer #1: All comments have been addressed

Reviewer #2: All comments have been addressed

2. Is the manuscript technically sound, and do the data support the conclusions?

Reviewer #1: Yes

Reviewer #2: Yes

3. Has the statistical analysis been performed appropriately and rigorously? 

Reviewer #1: Yes

Reviewer #2: Yes

4. Have the authors made all data underlying the findings in their manuscript fully available?

Reviewer #1: Yes

Reviewer #2: Yes

5. Is the manuscript presented in an intelligible fashion and written in standard English?

Reviewer #1: Yes

Reviewer #2: Yes

6. Review Comments to the Author

Reviewer #1: (No Response)

Reviewer #2: Authors have addressed all my queries and now the manuscript is technically sound. Therefore, I recommend this manuscript is accepted for publication in its current form.

7. PLOS authors have the option to publish the peer review history of their article (what does this mean?). If published, this will include your full peer review and any attached files.

Reviewer #1: No

Reviewer #2: No

---

## [Editor Report · Acceptance letter]

28 Feb 2022

PONE-D-21-31931R1 

Uterus-specific transcriptional regulation underlies eggshell pigment production in Japanese quail 

Dear Dr. Suzuki:

I'm pleased to inform you that your manuscript has been deemed suitable for publication in PLOS ONE. Congratulations! Your manuscript is now with our production department. 

Kind regards, 

on behalf of

Dr. Karthikeyan Adhimoolam 

Academic Editor

PLOS ONE